# Dynamic Compressive Behavior and Fracture Mechanisms of Binary Mineral Admixture-Modified Concrete

**DOI:** 10.3390/ma18122883

**Published:** 2025-06-18

**Authors:** Jianqing Bu, Qin Liu, Longwei Zhang, Shujie Li, Liping Zhang

**Affiliations:** 1Key Laboratory of Ministry of Education of Roads and Railway Engineering Safety Control, Shijiazhuang Tiedao University, Shijiazhuang 050043, China; bujq2004@163.com; 2School of Civil Engineering, Shijiazhuang Tiedao University, Shijiazhuang 050043, China; 1202401306@student.stdu.edu.cn (L.Z.); 1202401069@student.stdu.edu.cn (S.L.);

**Keywords:** mineral admixtures, dynamic mechanical properties, fracture mechanisms, three-dimensional mesoscale model

## Abstract

Fly ash and slag powder, as two of the most widely utilized industrial solid waste-based mineral admixtures, have demonstrated through extensive validation that their combined incorporation technology effectively enhances the mechanical properties and microstructural characteristics of concrete. Systematic investigations remain imperative regarding material response mechanisms under dynamic loading conditions. This study conducted microstructural analysis, static compression tests, and dynamic Split Hopkinson Pressure Bar (SHPB) impact compression tests on concrete specimens, complemented by dynamic impact simulations employing an established three-dimensional mesoscale concrete aggregate model. Through integrated analysis of macroscopic mechanical test results, mesoscale numerical simulations, and microstructural characterization data, the research systematically elucidated the influence mechanisms of different mineral admixture combinations on concrete’s dynamic mechanical behavior, energy dissipation characteristics, and fracture mechanisms. The results showed that all specimens exhibited strain rate enhancement characteristics as the strain rate increased. As the admixture approach transitioned from non-admixture to single admixture and subsequently to binary admixture, the dynamic strength, elastic modulus, and DIF of concrete increased progressively. Both the energy dissipation capacity and its proportion relative to total energy absorption showed continuous enhancement. The simulated stress–strain curves, failure modes, and fracture processes show good agreement with experimental results, this effectively verifies both the scientific validity of the mesoscale concrete model’s multiscale modeling approach and the reliability of the numerical simulations. Compared to FHC1, FMHC1’s mesoscale structure can more effectively convert externally applied energy into stored internal energy, thereby achieving superior dynamic compressive energy dissipation capacity.

## 1. Introduction

In current engineering practice of high-performance concrete systems, the synergistic application of high-efficiency chemical admixtures and special cementitious materials (such as mineral admixtures) emerged as a crucial technical approach to optimize performance limitations of high performance concrete [1]. As typical industrial solid waste-based supplementary cementitious materials, fly ash and slag powder demonstrated benefits through optimized design of cementitious material systems [2]. Their innovative application not only effectively reduced cement-based material consumption but also enhanced the comprehensive lifecycle performance of concrete [3]. Meanwhile, this approach demonstrated significant environmental benefits and promoted low-carbon development in the concrete industry.

Fly ash, as a pozzolanic material, exhibited hydraulic reactivity and underwent hydration reactions in alkaline environments to form cementitious substances [4,5]. When incorporated into cement-based materials, it induced three primary effects: the pozzolanic effect, micro-aggregate effect, and particle morphology effect [6]. These effects not only improved the workability of fresh concrete mixtures but also enhanced various hardened concrete properties. The hydration activity of SiO_2_ and Al_2_O_3_ in slag demonstrated intermediate characteristics between cement clinker and fly ash, showing latent hydraulicity that qualified it as a high-quality cement blending material and concrete admixture. The particle size differences among cement, fly ash, and slag enabled superior mutual filling in cementitious systems when dual incorporation was implemented, compared to single incorporation of either material. This optimized particle packing enhanced the compactness of hydrated cementitious matrices, reduced porosity in cement paste, and more effectively improved both mechanical properties and durability of concrete [7,8]. Furthermore, the dual incorporation strategy achieved complementary advantages by combining the respective merits of fly ash concrete and slag concrete while mitigating their individual limitations.

Current research on mineral admixture primarily focuses on two categories: single admixture and binary admixture system. In studies of single mineral admixture concrete, domestic and international scholars have conducted extensive investigations on fly ash and slag powder. Moghaddam et al. [9] discovered that finely ground fly ash increased cement hydration heat release, improved paste structure compactness and compressive strength, but reduced flowability. Cui et al. [10] confirmed through adjusted dosage and particle size distribution that fly ash decreased concrete porosity, promoted hydration reactions, and enhanced compressive strength across various curing ages. Dong et al. [11] demonstrated that milling treatment enabled fly ash to function as micro-aggregates during early hydration stages while intensifying pozzolanic effects at later stages, significantly improving full-age concrete strength. Luo et al. [12] revealed that increased slag content reduced initial fluidity and setting time of cement paste, while finer particle size shortened initial setting time and improved mechanical properties. Yang et al. [13] achieved 27% and 15% strength improvements at 3d and 28d, respectively, in eco-friendly ultra-high performance concrete with 40% slag incorporation, accompanied by refined pore structures. Cheng et al. [14] conducted mechanical tests on hybrid fiber-reinforced fly ash concrete, finding that fiber and fly ash incorporation effectively inhibited microcrack propagation and enhanced overall mechanical performance. In binary mineral admixture concrete research, scholars achieved superior results through complementary utilization of different mineral admixtures. Yu et al. [15] developed a composite mineral powder containing slag-silica fume-titanium slag, which reduced cement consumption by 40–60% with chemical admixtures. Although early strength slightly decreased, significant later-stage strength development was observed. Duan et al. [16] created a slag-fly ash-desulfurization gypsum composite activator that replaced 75–95% cement, producing mortar with high compressive strength, low hydration heat, and reduced porosity. Hu et al. [17] reported that higher single fly ash dosage decreased 7d and 28d concrete strength, while fly ash-slag combination showed limited 7d strength improvement but significantly enhanced 28d strength, along with improved slump and reduced plastic viscosity. Huang et al. [18] identified optimal comprehensive mechanical properties in cement mortar with 20% fly ash and 30% slag through proportion optimization.

Furthermore, the presence of coarse aggregates in concrete materials resulted in differentiated crack propagation paths and crack resistance characteristics under dynamic loading conditions. Research on dynamic mechanical properties of modified concrete materials held significant implications for designing impact-resistant structures and special load-bearing systems. Feng et al. [19] proposed that coarse aggregates played a crucial role in compressive strength enhancement, noting distinct compressive strength variations in concrete synthesized with different fly ash activators. The activator dosage was found to significantly influence concrete’s strain rate sensitivity. Lian et al. [20] demonstrated that fly ash and slag based alkali-activated concrete exhibited notable strain rate effects in both compression and tension, with strength grades substantially affecting its strain rate sensitivity. Zou et al. [21] conducted dynamic mechanical studies on ultrafine slag powder cement paste (USP-CP), revealing that the molar ratio of elemental components in raw materials exerted more pronounced influence on dynamic compressive strength than static strength. Excessive introduction of Na and Si elements in alkaline activators was observed to suppress the dynamic mechanical enhancement of USP-CP.

From a mesoscopic perspective, SAC was conceptualized as a three-phase composite material comprising cement mortar, coarse aggregates, and the ITZ between them. Since the concept of “mesoscale concrete modeling” was proposed, researchers have achieved realistic simulations of failure characteristics in heterogeneous concrete through mesoscale aggregate models. Chen et al. [22] established a mesoscale model demonstrating high reliability in analyzing concrete strength and failure patterns. Guo et al. [23] treated concrete as a two-phase composite of spherical granite aggregates and mortar, investigating granite’s reinforcement mechanism from a mesoscopic perspective. Xu et al. [24] developed a mesoscale numerical model incorporating matrix, aggregates, and ITZ, effectively explaining damage evolution processes in concrete materials under different strain rates. Li et al. [25] created a polyhedral aggregate concrete model at mesoscale, numerically verifying strain rate effects on energy transfer and internal damage progression.

In summary, the composite blending approach demonstrated the capability to combine the advantages of both fly ash concrete and slag concrete, proving to be a practical and effective modification method for significantly enhancing concrete mechanical properties. However, current research on dual fly ash-slag concrete exhibits insufficient depth in mechanical performance characterization with limited experimental datasets. Furthermore, studies investigating the effects of single/dual incorporation of fly ash and slag on concrete dynamic mechanical responses remain scarce. To address these limitations, this study conducted microstructural analysis, static compression tests, and dynamic compression experiments on plain concrete (HC), single mineral admixture concrete (FHC) and binary mineral admixture concrete (FMHC). Dynamic impact simulations were performed using established 3D mesoscale concrete aggregate models to analyze damage evolution patterns in aggregates, mortar, and ITZ, with simulation results cross-validated against experimental findings. Through integrated analysis of macro-mechanical test results, mesoscale simulations, and microscopic characterization data, the influence mechanisms of different mineral admixture combinations on concrete dynamic mechanical behavior, energy dissipation characteristics, and failure modes were systematically elucidated.

## 2. Experimental Investigation

### 2.1. Specimen Preparation

The cementitious material system used in this study includes cement, fly ash, and slag powder (as shown in Figure 1). P.O42.5 grade ordinary silicate cement was used for cement. The basic physical parameters of fly ash and slag powder are as follows: density of 2200 kg/m^3^, 2840 kg/m^3^, bulk density of 930 kg/m^3^, 1000 kg/m^3^, specific surface area of 450 m^2^/kg, 500 m^2^/kg, and the chemical composition is shown in Table 1. Coarse aggregate adopts 2~5 mm, 5~8 mm, 8~11 mm three levels of continuous grading gravel, its basic performance indicators are shown in Table 2; fine aggregate for natural river sand, mud content of 1.8%, fineness modulus 2.4. Concrete ratio design follows the ‘Technical specification for high performance concrete’ (GB/T 41054-2021) [26] design principles. In the total cementitious materials, the dosage of fly ash is set to be 20% and 40%, and the dosage of slag powder is set to be 10% and 20%, and the proportion of total mineral admixture replacing cement is fixed at 40%. This mixing range is based on the following research bases: when mixing mineral admixture alone, 20%~40% of cementitious materials can effectively improve the working performance of concrete [27]; in the compound mixing system, slag powder can significantly improve the compactness and interfacial bonding performance of cementitious materials when mixed with about 20% of slag powder [28]. The final mix designs of five high-performance concrete formulations are presented in Table 3.

Static specimens were prepared in accordance with the Standard for Test Methods of Physical and Mechanical Properties of Concrete (GB/T 50081-2019) [29], following standardized procedures for quasi-static compression and split tensile testing. Cubic concrete specimens with a side length of 100 mm were prepared using a triple mold of the size 100 mm×100 mm×100 mm, and cured under standard curing conditions until the age of 28d. The size of the steel mold for the fresh concrete casting is 300 mm×300 mm×300 mm, the preparation of the size of the concrete specimen is 300 mm×300 mm×150 mm, standard curing 28d after taking the core, cutting, grinding and other processes, processed into a diameter of 75 mm, 37.5 mm high cylindrical dynamics specimen, and to ensure that the specimen end face flatness ≤ 0.02 mm, the top and bottom of the end face of the degree of non-parallelism is not greater than 1% of the diameter. In order to facilitate the data analysis of the subsequent dynamic experiments. Some of the prepared static and dynamic specimens are shown in Figure 2.

### 2.2. Experimental Method

Static mechanical tests were conducted using an MYE-30008 computer-controlled electro-hydraulic servo pressure testing machine made by China Zhongzhong Heavy Industry Science and Technology Company (Changsha, China). As shown in Figure 3a, the installation of specimens for static compression tests followed standard loading procedures: first, specimens were pre-compressed to eliminate gaps in the loading system and ensure uniform contact between the specimen and loading plates; then, displacement-controlled loading was implemented through the control system according to preset protocols until specimen failure. The loading process used a constant strain rate control of $1\times 10^{-5}\;s$, while simultaneously collecting force–displacement curve data. The quasi-static compressive strength of the specimens was taken as the arithmetic mean of three valid test results, with abnormal data excluded to ensure testing accuracy.

The static splitting test was based on the splitting principle of the Brazilian disc method, with loading pads added to the compression test device: as shown in Figure 3b, steel pads with a width of 5mm were symmetrically placed at the center of the top and bottom surfaces of the specimen. The ratio of pad width to cube edge length was 4% (meeting the dimensional effect negligibility conditions proposed in references [30]), ensuring that the load formed a linearly distributed splitting tensile stress through the pads, conforming to the plane stress assumption in elastic mechanics for splitting tests. The loading control method was consistent with the compression tests, using displacement control mode with a loading strain rate of $1\times 10^{-6}\;s$, and calculating the splitting tensile strength using elastic fracture mechanics formulas. This pad design effectively avoided unintended failure due to stress concentration, ensuring that the specimen split along the preset vertical plane, thereby obtaining reliable tensile strength indicators.

The SHPB test system is a classic apparatus for studying the dynamic mechanical properties of quasi-brittle materials such as rock and concrete, effectively simulating material response behavior under high strain rates from explosive impact loads. This test used a Φ75 mm SHPB device (as shown in Figure 4), which mainly consists of a launching power unit, incident bar, transmission bar, absorption bar, and dynamic strain measurement system. The bar material was high-strength alloy steel with a density of 7850 kg/m^3^, elastic modulus of 210 GPa, and Poisson’s ratio of 0.3, resulting in a calculated longitudinal wave velocity of 5173 m/s. Both the incident and transmission bars were 2 m in length, satisfying the long bar assumption conditions of one-dimensional stress wave theory. The dynamic loading system employed an improved spindle-shaped striker, generating a half-sine wave incident stress wave through pneumatic drive. When the stress wave propagated along the incident bar to the specimen interface, part of the energy formed a reflected wave returning to the incident bar, while the other part passed through the specimen forming a transmitted wave entering the transmission bar. During signal acquisition, resistance strain gauges (CH1, CH2) were symmetrically attached to the middle of the incident and transmission bars, and strain signals were synchronously collected through an ultra-dynamic strain gauge. After wave processing, the time–history curves of incident, reflected, and transmitted waves were obtained. Through the “three-wave method,” [31] the average stress, $\sigma_{s}\left(t\right)$ average strain $\varepsilon_{s}\left(t\right)$, and average strain rate of the concrete specimens ${\dot{\varepsilon }}_{s}\left(t\right)$ were obtained, with calculation formulas as shown in Equations (1)–(3):(1)$$\sigma_{s}\left(t\right)=\frac{A_{0}E_{0}}{2A_{s}}\left[\varepsilon_{I}\left(t\right)+\varepsilon_{R}\left(t\right)+\varepsilon_{T}\left(t\right)\right]$$(2)$$\varepsilon_{s}\left(t\right)=\frac{C_{0}}{L_{s}}\int_{0}^{t}\left[\varepsilon_{I}\left(t\right)-\varepsilon_{R}\left(t\right)-\varepsilon_{T}\left(t\right)\right]dt$$(3)$${\dot{\varepsilon }}_{s}\left(t\right)=\frac{C_{0}}{L_{s}}\left[\varepsilon_{I}\left(t\right)-\varepsilon_{R}\left(t\right)-\varepsilon_{T}\left(t\right)\right]$$
where $\varepsilon_{I}\left(t\right)$, $\varepsilon_{R}\left(t\right)$, and $\varepsilon_{T}\left(t\right)$ represent the incident, reflected, and transmitted strains, respectively; $A_{0}$, $E_{0}$, and $C_{0}$ represent the cross-sectional area, modulus of elasticity, and longitudinal wave velocity of the bar, respectively; $A_{s}$ and $L_{s}$ represent the cross-sectional area and length of the concrete specimen, respectively.

The stress uniformity of the specimen during loading is a prerequisite for the validity of the test [32]. The strain–time course curve under typical impact air pressure is selected for balance verification, as shown in Figure 5. The sum of the strain signals from the incident wave and reflected wave overlaps well with the transmitted wave strain signals, demonstrating that the specimen is dynamically balanced at both ends [33]. This observation suggests that the experimental results are valid.

The tensile failure process of concrete structures under dynamic disturbances during service is also a process of energy conversion [34]. Analyzing concrete material failure from an energy perspective better reflects the damage evolution patterns of specimens under different working conditions. Based on the characteristics of dynamic impact tests, dissipated energy is closely related to tensile damage and can serve as an evaluation index for concrete material toughness. Based on the law of conservation of energy [35], the energy carried by incident waves, reflected waves, transmitted waves during the entire impact process, and the energy dissipated by specimen failure can be obtained by Equations (4)~(7), and the energy absorption rate η can be obtained by Equation (8):(4)$$W_{I}(t)=C_{0}A_{b}E_{b}\int_{0}^{t}\overset{2}{\varepsilon_{I}(t)}dt$$(5)$$W_{R}(t)=C_{0}A_{b}E_{b}\int_{0}^{t}\overset{2}{\varepsilon_{R}(t)}dt$$(6)$$W_{T}(t)=C_{0}A_{b}E_{b}\int_{0}^{t}\overset{2}{\varepsilon_{T}(t)}dt$$(7)$$W_{S}(t)=W_{I}(t)-W_{R}(t)-W_{T}(t)$$(8)$$\eta =\frac{W_{S}}{W_{I}}$$
where $W_{I}(t)$ is the incident wave energy; $W_{R}(t)$ is the reflected wave energy; $W_{T}(t)$ is the transmitted wave energy; $W_{S}(t)$ is the energy dissipated by the specimen; $C_{0}$ is the longitudinal wave velocity of the pressure bar; $A_{b}$ and $C_{0}$ are the cross-sectional area and elastic modulus of the pressure bar, respectively; t represents any moment in time.

## 3. Experimental Results

### 3.1. Microstructure

XRD tests were carried out using an X-pert 3-powder X-ray powder diffractometer, and then the physical phase was retrieved and analyzed using the X-pert Highscore software (https://www.malvernpanalytical.com/en/products/category/software/x-ray-diffraction-software/highscore, accessed on 10 January 2025) accompanying the X-ray powder diffractometer. SEM scanning tests were carried out using a VEGA-3S-BH tungsten filament scanning electron microscope made by China QiaoBang Instrumnet (Lishui, China).

Comparing the microstructural composition of HC, FHC1, and FHC2 (Figure 6a), the content of quartz and mullite shows a progressive upward trend. This is primarily due to the characteristics of fly ash, where during the hydration and hardening process of concrete, some alumino-silicate compounds in fly ash gradually transform to form mullite phases. Meanwhile, the diffraction peak intensity of calcium hydroxide significantly decreases because fly ash reacts with calcium hydroxide in a secondary hydration reaction. Calcium hydroxide, as a key reactant, is largely consumed to produce more stable products such as calcium silicate hydrate (C-S-H) gel [36]. When both fly ash and slag powder are simultaneously added to concrete, the diffraction peak intensity of dicalcium silicate (C_2_S) further increases. This occurs because slag powder works synergistically with fly ash, with its active components participating in secondary hydration, providing additional silicon and aluminum components, further optimizing the formation and structure of C-S-H gel [37].

As shown in Figure 6b, with the sole addition of fly ash, as the fly ash content increases, the number of pores and microcracks in the concrete gradually decreases. This is because fly ash actively participates in the hydration reaction of cement, promoting the formation of large amounts of C-S-H gel. Calcium hydroxide, as a reactant, is continuously consumed, creating conditions for the continued formation of C-S-H gel, causing the originally loose structure of concrete to gradually become compact, thereby effectively improving the overall strength and hardness of the concrete. When both fly ash and slag powder are simultaneously added to concrete, compared to the addition of fly ash alone, the number of pores and microcracks further decreases, and structural density is more significantly enhanced [38]. This is because slag powder works synergistically with fly ash, accelerating the hydration process of cement and promoting the rapid formation of more C-S-H gel [39]. The bonding areas between aggregates and cementitious materials increase, significantly expanding the toughness range of concrete and greatly enhancing the concrete’s resistance to cracking and deformation.

### 3.2. Static Test Results

Figure 7 shows the test results of axial compressive strength and splitting tensile strength of concrete with different mineral admixture combinations. As mineral admixtures transition from none to single admixture, and then to binary admixture, concrete specimens exhibit higher compressive and splitting resistance performance. Compared with HC, the average axial compressive strength of FHC1, FHC2, FMHC1, and FMHC2 increased by 10.11%, 11.75%, 18.63%, and 25.52% respectively, while the average splitting tensile strength increased by 1.40%, 5.61%, 11.58%, and 18.95%, respectively. These strength enhancement characteristics are closely related to the synergistic effect of composite mineral admixtures, where active components improve matrix density through micro-aggregate filling and pozzolanic reactions, optimizing the bonding performance of the ITZ, thereby effectively enhancing the mechanical properties of concrete.

### 3.3. Dynamics Test Results

#### 3.3.1. Dynamic Stress–Strain Curves

Figure 8 shows the stress–strain curves of each specimen under 0.3–0.5 MPa impact air pressure. Under the fixed impact load condition, the peak stresses of the specimens showed a significant increasing trend as the admixture approach transitions from non-admixture to single admixture, and further to binary admixture. Specifically, under the impact load of 0.3 MPa, the peak stresses of the specimens with different combinations of mineral dopants were 37.08 MPa, 40.05 MPa, 43.41 MPa, 46.92 MPa, and 48.31 MPa, and the strengths of the specimens with different combinations of mineral dopants were 8.01%, 17.07%, 26.54%, and 30.28%, respectively, compared with those of the HC specimens. FHC1, FHC2, FMHC1, and FMHC2 specimens, respectively, compared with the peak stress of HC specimens. The linear elastic phase characteristics of the stress–strain curves indicated that the dynamic modulus of elasticity showed a significant increasing trend with the improvement of the mechanical properties of concrete, with the steepness of the curve for the specimens with compounded mineral admixtures being particularly obvious. Under the influence of different impact loads, the peak stress increases with the increase in impact load. Taking HC specimens as an example, the peak stresses increased by 4.75% and 14.27% in the impact load variation interval from 0.3 to 0.5 MPa, respectively. When the impact load reaches 0.4MPa, the plastic deformation stage of each specimen begins to show a small expansion; when the load is further increased to 0.5MPa, the time for the specimen to reach the peak stress is significantly shortened, and the stress–strain curve rapidly transitions from the elastic stage to the plastic flow stage.

Figure 9 presents the variation curves of dynamic compressive strength and Dynamic Increase Factor (DIF) for concrete with different mineral admixture combinations. Analysis reveals that the dynamic compressive strength of all specimens shows significant strain rate sensitivity, and concrete specimens containing mineral admixtures have higher DIF values. As mineral admixtures transition from none to single to binary admixture, the dynamic compressive strength of concrete shows a continuous growth trend, and its resistance to impact loads is optimized accordingly. This indicates that when fly ash and slag powder are added together to concrete, they promote a denser internal structure, which can effectively suppress crack initiation and propagation in the ITZ region, enabling the material to exhibit stronger energy dissipation capacity and deformation bearing capacity under high strain rates [20]. Comparing the DIF curves of different admixture systems, the enhancement in dynamic strength of concrete with composite mineral admixtures is significantly higher than that of single admixture systems, indicating a synergistic effect between fly ash and slag powder in improving the strain rate response characteristics of concrete.

Figure 10 shows the variation curves of dynamic elastic modulus with different mineral admixture combinations. When impact air pressure increases from 0.3 MPa to 0.5 MPa, the dynamic elastic modulus of concrete specimens generally shows an upward trend. Under higher impact air pressures, concrete materials exhibit strain rate effects during impact wave transmission, thereby promoting an increase in dynamic elastic modulus. Under the same impact air pressure, as mineral admixtures transition from none to single to binary admixture, the dynamic elastic modulus of concrete generally shows an upward trend. This phenomenon is attributed to the optimization effect of composite mineral admixture systems on the mesoscopic interface properties of concrete, reducing the porosity of ITZ and enhancing the bonding strength between aggregates and mortar matrix. When subjected to impact loads, this strengthened mesoscopic interface structure can more effectively suppress the debonding and sliding between matrix and aggregate phases, reducing energy dissipation loss at interfaces, allowing the material to maintain higher stiffness response during the elastic deformation phase, ultimately manifesting as a significant enhancement in dynamic elastic modulus.

#### 3.3.2. Energy Characterization

Based on Equations (4)~(8), the energy characterization is calculated for different working conditions, and the results of energy characterization are shown in Table 4.

As shown in Figure 11a, under fixed impact loading conditions, as mineral admixtures transition from none to single to binary admixture, the dissipated energy of specimens shows a significant increasing trend. This is because the composite system of mineral admixtures significantly enhances the dynamic compressive strength and energy storage capacity of concrete by optimizing ITZ bonding performance and matrix density. When impact loads act, binary mineral admixture concrete can withstand dynamic stresses of higher amplitude and dissipate more energy through plastic deformation and microcrack propagation before reaching the failure threshold, resulting in an increasing trend of dissipated energy proportion with the degree of binary admixture.

Figure 11b reveals the influence pattern of different impact loads on the proportion of dissipated energy: the proportions of dissipated energy for HC under impact loads of 0.3 MPa, 0.4 MPa, and 0.5 MPa are 15.25%, 16.71%, and 17.44% respectively, while the corresponding proportions for FMHC2 significantly increase to 38.40%, 40.75%, and 45.86%. The lower proportion of dissipated energy for HC under low load conditions stems from the stress concentration effect of its internal initial defects, leading to premature material failure and inability to effectively bear impact energy; under high load conditions, the crack propagation speed exceeds the energy dissipation rate, causing the growth rate of dissipated energy proportion to become moderate with increasing load. In contrast, FMHC2, with its excellent dynamic ductility, fully absorbs energy through the coordinated deformation of mortar matrix and ITZ during low-speed impact; when impact speed increases, the activation threshold of its internal energy dissipation mechanism expands, allowing the proportion of dissipated energy to maintain a significant growth trend over a wider load range.

#### 3.3.3. Damage Morphology

Figure 12 illustrates the failure pattern characteristics of specimens incorporating diverse mineral admixture combinations under impact compressive loading. Under fixed-impact loading conditions, as mineral admixtures transition from absence to single and further to binary incorporation, the impact resistance of the specimens exhibits a progressive strengthening trend. At an impact load of 0.3 MPa, localized spalling damage is observed in FHC1 specimens, whereas FMHC1 specimens retain superior structural integrity. When the impact load increases to 0.5 MPa, FHC1 specimens undergo typical pulverization failure, leaving no significant large remnants post-fragmentation. In contrast, FMHC1 specimens produce fragments with dimensions substantially larger than those of FHC1 specimens, preserving a higher proportion of intact fragments. Failure mechanism analysis reveals that HC specimens, characterized by elevated internal porosity, develop stress concentration effects under impact loading. This phenomenon promotes preferential crack propagation along porous defects, ultimately forming penetrating primary cracks that trigger material instability. Conversely, FMHC specimens subjected to dynamic loading demonstrate rapid internal energy release, resulting in abrupt brittle fracture. This failure mode manifests as straight-line main crack propagation patterns that traverse the specimen cross-section. These observations indicate that composite mineral admixture systems effectively mitigate crack penetration and fragmentation tendencies under impact loading. This enhancement is attributed to improved matrix densification and interfacial bonding performance, thereby bolstering concrete’s resistance to impact-induced damage.

## 4. Numerical Simulation

### 4.1. Modeling of Mesoscale Aggregates

This section uses the Monte Carlo algorithm to place random aggregates at random positions in a predetermined space, with penetration judgment between aggregates to ensure they do not intersect in the spatial region. Random polyhedral aggregates can characterize the irregular features of concrete coarse aggregates, effectively reproducing the actual mesoscopic structure. The Fuller gradation function is used to determine the gradation relationship of aggregates in the model [40], which can achieve optimal particle size distribution characteristics in the concrete model, thereby achieving higher density. The gradation function is shown in Equation (9).(9)$$P\left(d\right)=100{\left(\frac{d}{d_{\max}}\right)}^{n}$$
where $P\left(d\right)$ represents the percentage of aggregate with particle size smaller than d; $d_{\max}$ is the maximum aggregate size; n is an empirical parameter. Following the aggregate gradation determination method in the reference [41], a gradation curve where n=0.5 is selected to determine aggregate content, as shown in Figure 13. Continuous gradation can be divided into different particle size grade ranges, and the proportion of coarse aggregates in different grade ranges can be calculated using Equation (10).(10)$$P\left[d_{s},d_{s+1}\right]=\frac{P(d_{s+1})-P(d_{s})}{P(d_{\max})-P(d_{\min})}$$
where $P\left[d_{s},d_{s+1}\right]$ represents the cumulative percentage of aggregates between particle size grade segments $\left[d_{s},d_{s+1}\right]$; $d_{\min}$ represents the diameter of the smallest coarse aggregate.

Based on the mapping projection meshing algorithm [42], the mesh generation process for the mesoscale concrete model is implemented through the following procedures: Initially, on the ANSYS 19.0 preprocessing platform, the explicit dynamic element Solid164 is employed to discretize the concrete specimen into hexahedral mapped meshes, establishing a homogenized grid model that is subsequently exported in K-file format compatible with LS-DYNA. Subsequently, within the Visual Studio integrated development environment, a Fortran program is developed to synergistically process the homogenized mesh K-file and the data file containing aggregate particle sizes and centroid coordinates. The computational workflow comprises two principal phases: First, adaptive adjustment of node/element numbering formats in the original data files is performed according to Fortran’s array indexing rules. Second, a meso-component identification algorithm is implemented to project geometric definitions of aggregates, mortar, and interfacial transition zones onto the predefined homogenized mesh model through point-by-point determination of element material attributes. Figure 14 illustrates the mesoscale aggregate concrete model constructed using three-dimensional mapping meshing technology. As demonstrated in the figure, the model exhibits excellent mesh uniformity, effectively mitigating potential simulation inaccuracies caused by mesh quality issues.

Following the construction of the concrete mesoscopic model, a three-dimensional finite element model of the incident and transmission bars is established using the LS-DYNA preprocessing platform LS-PREPOST. The pressure bars are geometrically modeled following mesh homogenization principles, with 20 mm equidistant elements along the longitudinal axis and 7 mm annular mesh discretization in radial cross-sections. The completed numerical model for the SHPB dynamic compression test is presented in Figure 15. To ensure work equivalence between numerical simulations and physical experiments, dynamic load input technology is employed to simulate projectile impact processes. The implementation involves three sequential operations: First, the experimentally measured incident stress time–history curve is converted into a *.txt format data file. Then, the Load_xy data curve is defined through the LS-DYNA keyword *Curve. Finally, the *Load_segment_set command is employed to precisely map the load curve onto the loading end-face of the incident bar. For the contact algorithm, the *CONTACT_ERODING_SURFACE_TO_SURFACE algorithm is implemented, with the concrete specimen designated as the slave contact surface and the pressure bar as the master contact surface. This configuration accurately simulates interface separation behavior following material failure during impact loading. At the transmission bar termination, non-reflective boundary conditions are applied to replicate the stress wave dissipation characteristics of the absorption bar, thereby ensuring the validity of the one-dimensional stress wave assumption under impact loading conditions.

### 4.2. Determination of Material Parameters

Based on the basic mechanical parameters of aggregates (Table 2) and the RHT model parameter calculation method proposed in Reference [43], the RHT model parameters of aggregates were initially determined through theoretical derivation (Table 5). For the mortar matrix material model of HC specimens, key mechanical parameters were established according to experimental data from Reference [44]. The parameters included a density of 2300 kg/m^3^, quasi-static compressive strength of 30.50 MPa, and Poisson’s ratio of 0.21. Referring to related research results, the compressive strength and elastic modulus of ITZ are usually 80%~85% of the mortar matrix [45]. Accordingly, this study takes 80% of the mortar performance indicators as the parameter benchmark for the ITZ material model, determining its density as 2300 kg/m^3^, quasi-static compressive strength 24.40 MPa, and Poisson’s ratio 0.21.

Based on these preliminary parameters, through parameter calibration and iterative optimization in numerical simulation, the mortar and ITZ material parameters applicable to concrete with different mineral admixture combinations were finally obtained (Table 6). This parameter optimization process ensures that the model can accurately characterize the energy dissipation characteristics and failure evolution laws of concrete mesoscopic components through dynamic mechanical response fitting.

### 4.3. Validation of Numerical Model of SHPB

Taking FHC1 and FMHC1 as examples, Figure 16 presents the stress–strain curves obtained from dynamic impact compression simulations and their comparison with experimental results. The analysis shows good agreement between numerical simulations and experimental measurements in terms of stress–strain behavior. Compared to FHC1, FMHC1 exhibits higher dynamic compressive strength under identical impact loading conditions. This difference indicates that FMHC1 possesses superior mechanical properties.

Figure 17 illustrates the typical failure process of specimens under dynamic impact compression simulation. Comparative analysis demonstrates good agreement between the simulated and experimental failure processes. The established mesoscopic concrete model effectively simulates both dynamic mechanical properties and failure characteristics of concrete specimens. These findings confirm the feasibility of the mesoscopic concrete modeling method and validate the numerical simulation results.

### 4.4. Simulation Results

#### 4.4.1. Damage Morphology

Figure 18 illustrates the failure patterns of specimens under impact compressive loading. All specimens demonstrate characteristic radial tensile failure, exhibiting multiple vertical and oblique cracks radiating along the loading axis. These cracks progressively propagate with distinct brittle fracture features. Analysis of mesoscale structure damage reveals more pronounced degradation in the mortar matrix and ITZ compared to relatively minor damage in aggregates. This phenomenon originates from the ITZ’s inferior mechanical strength and toughness relative to the mortar matrix and aggregates, leading to stress concentration that initiates cracking in these interfacial regions. Comparative analysis of damage morphology reveals that HC specimens exhibit higher crack development capacity, while FMHC1 specimens demonstrate limited crack development. This contrast confirms the effectiveness of binary mineral admixture in enhancing concrete impact resistance. Experimental observations align with numerical simulation results, validating the capability of the mesoscale model in characterizing concrete failure mechanisms under impact loading.

#### 4.4.2. Energy Characterization

Comparative analysis of energy time–history curves for meso-scale structures in FHC1 and FMHC1 specimens (Figure 19) yields the following conclusions: At 0.4 ms, the incident stress wave reaches the specimen boundary, activating the material’s energy absorption mechanism. Analysis of internal energy evolution curves demonstrates that the mortar matrix attains the highest internal energy peak, followed by the ITZ, with aggregate particles exhibiting the lowest values. Kinetic energy time–history curves reveal analogous trends: the mortar matrix displays the most pronounced kinetic energy response, succeeded by aggregate particles, while the ITZ shows minimal kinetic energy increment. These findings confirm that the mortar matrix and ITZ act as primary energy dissipation carriers under dynamic loading. Their distinct energy absorption characteristics dictate crack propagation paths predominantly concentrated in these regions. In contrast, aggregate particles exhibit weaker internal energy absorption capacity, corresponding to relatively limited damage development.

Further analysis demonstrates that FHC1 specimens exhibit significantly lower total internal energy absorption compared to FMHC1 specimens while displaying reversed superiority in total kinetic energy absorption. Integration with mesoscale structures’ energy dissipation pattern reveals that FMHC1’s mesoscale structures more effectively convert externally input energy into internal energy storage than FHC1, thereby reducing kinetic energy dissipation losses and optimizing dynamic compression energy dissipation capacity. These energy characteristics confirm macroscopic experimental observations, ultimately validating the enhancement effect of binary mineral admixture on concrete’s dynamic mechanical performance.

## 5. Discussion

The experimental and numerical findings of this study deepen and expand the current understanding of mineral admixture optimization in concrete materials. Existing studies demonstrate that fly ash enhances particle packing density through its spherical particle morphology [36], while slag powder improves pore structure by promoting secondary hydration reactions [37]. The binary combination exhibits significant synergistic effects, leading to further optimization of mechanical properties. Specifically, compared with the reference group (HC), the FMHC2 group shows a 25.52% increase in uniaxial compressive strength. This improvement outperforms traditional single mineral admixture systems (consistent with conclusions from references [15,16]), revealing optimization mechanisms of performance in mortar and ITZ, which aligns with microstructural test results in reference [38]. Notably, the DIF of specimens shows progressive enhancement from HC to FMHC groups, consistent with findings in reference [20].

Quantitative analysis of mesoscale structure energy distribution through simulation reveals the following hierarchy: mortar matrix > ITZ > aggregate. This conclusion explains how mortar and ITZ structures effectively alleviate stress concentration at aggregate interfaces under dynamic loading conditions while identifying key failure mechanisms [31]. Damage evolution analysis of concrete cross-sections demonstrates that under impact loading (Figure 20), specimen damage initiates with stress concentration at weak aggregate interfaces. Initial damage occurs in ITZ regions and then propagates along loading directions. Continuous cracks form when ITZ connects with mortar [24]. The binary mineral admixture effectively enhances bonding performance between aggregates and mortar in concrete, improving impact resistance and validating the core findings of this study.

## 6. Conclusions

In this paper, microstructural analysis, static compression tests, and dynamic compression experiments were conducted on HC, FHC, and FMHC. Dynamic impact simulations were carried out using an established three-dimensional mesoscale concrete aggregate model, and the simulation results were cross-validated with the experimental results. The results of this study concluded as follows:(1)The synergistic effect between slag powder and fly ash further optimizes the gel system structure, thereby comprehensively enhancing the stability of concrete’s internal microstructure. Concrete specimens incorporating binary mineral admixture demonstrate higher peak stresses in static compression tests and exhibit greater slopes during the elastic phase of splitting stress–strain curves. Compared to HC, the FHC1, FHC2, FMHC1, and FMHC2 specimens show respective increases of 10.11%, 11.75%, 18.63%, and 25.52% in average axial compressive strength, along with corresponding improvements of 1.40%, 5.61%, 11.58%, and 18.95% in average splitting tensile strength.(2)All specimens exhibited strain rate enhancement characteristics as the strain rate increased. As the admixture approach transitioned from non-admixture to single admixture and subsequently to binary admixture, the dynamic strength, elastic modulus, and DIF of concrete increased progressively. Both the energy dissipation capacity and its proportion relative to total energy absorption showed continuous enhancement.(3)Comparative analysis demonstrates that the simulated stress–strain curves, failure modes, and fracture processes show good agreement with experimental results; this effectively verifies both the scientific validity of the mesoscale concrete model’s multiscale modeling approach and the reliability of the numerical simulations. The findings establish a robust basis for investigating critical micromechanical behaviors of concrete materials, including damage accumulation mechanisms and dynamic crack propagation laws under complex multi-field coupling conditions.(4)From an energy perspective, the mortar matrix exhibits the highest internal energy peaks, followed by the ITZ, with aggregate particles showing the lowest values. The mortar matrix and ITZ serve as primary contributors to energy dissipation under dynamic loading, whereas aggregate particles demonstrate relatively lower damage levels due to their weak internal energy absorption capacity. Compared to FHC1, FMHC1’s mesoscale structure can more effectively convert externally applied energy into stored internal energy while minimizing kinetic energy losses, thereby achieving superior dynamic compressive energy dissipation capacity.

## Figures and Tables

**Figure 1 materials-18-02883-f001:**
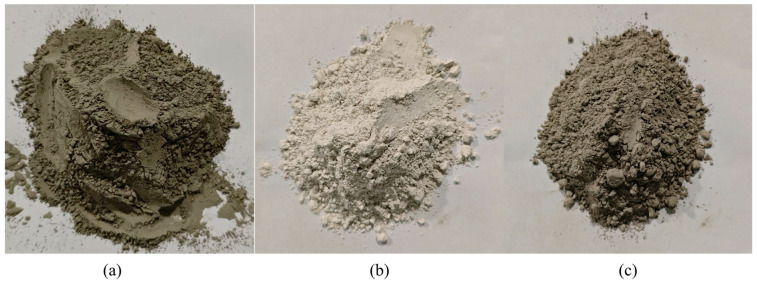
Cementitious materials. (**a**) Cement; (**b**) slag powder; (**c**) fly ash.

**Figure 2 materials-18-02883-f002:**
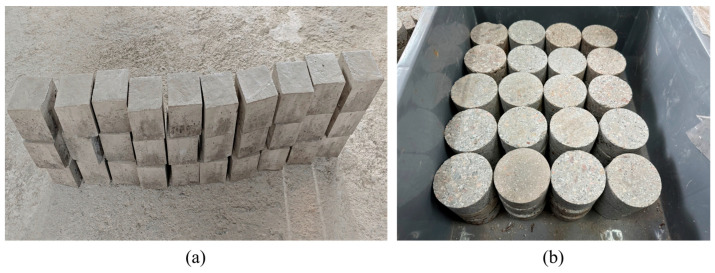
Partial static and dynamic specimens. (**a**) Static test specimen; (**b**) dynamic test specimen.

**Figure 3 materials-18-02883-f003:**
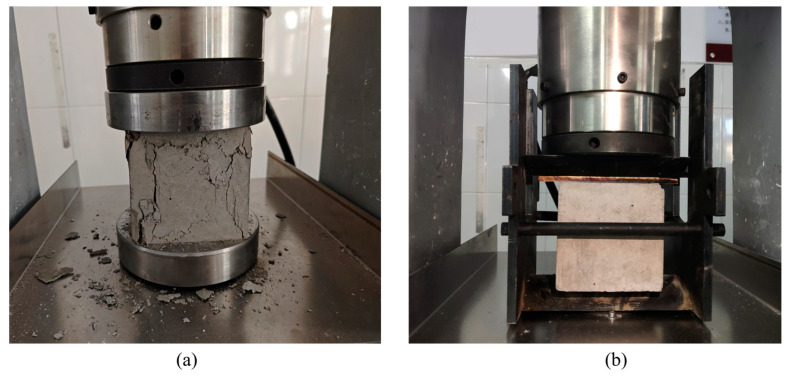
Static mechanics experimental setup. (**a**) Static uniaxial compressive test; (**b**) static split tensile test.

**Figure 4 materials-18-02883-f004:**
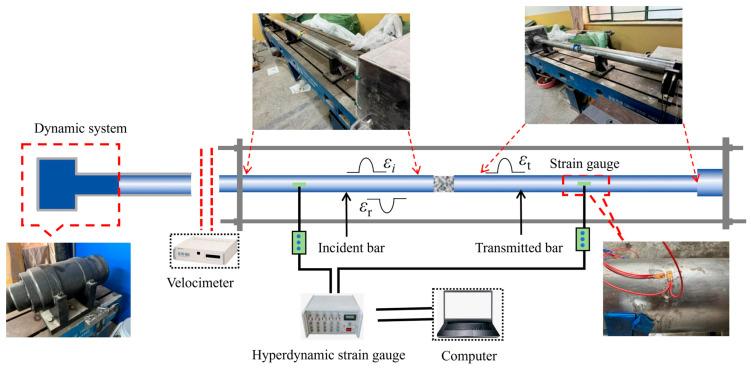
SHPB test system.

**Figure 5 materials-18-02883-f005:**
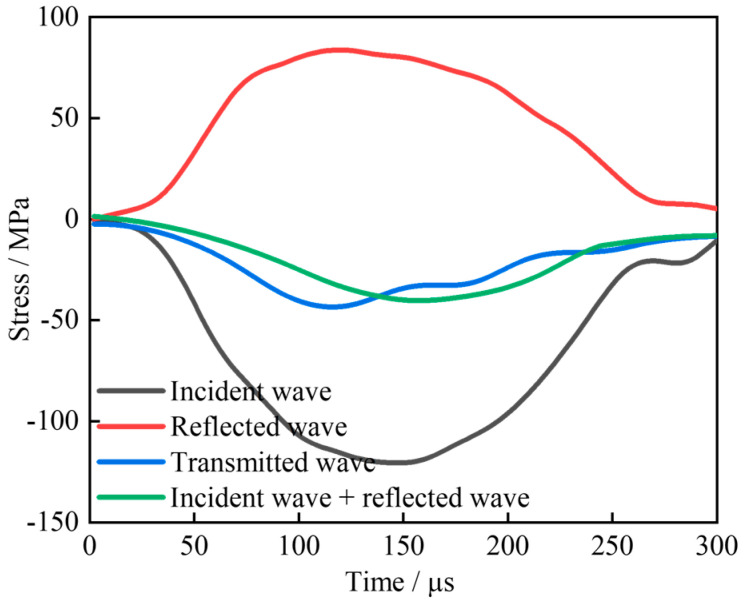
Dynamic balance verification.

**Figure 6 materials-18-02883-f006:**
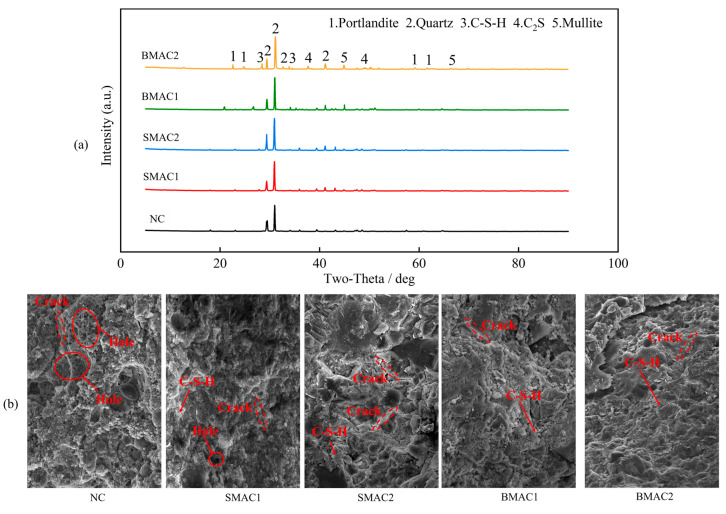
Microscopic characteristics of concrete. (**a**) Elemental composition; (**b**) microscopic morphology.

**Figure 7 materials-18-02883-f007:**
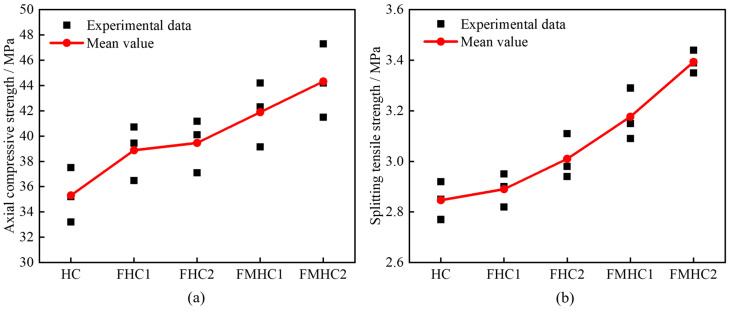
Test results of static tests. (**a**) Axial compressive strength; (**b**) splitting tensile strength.

**Figure 8 materials-18-02883-f008:**
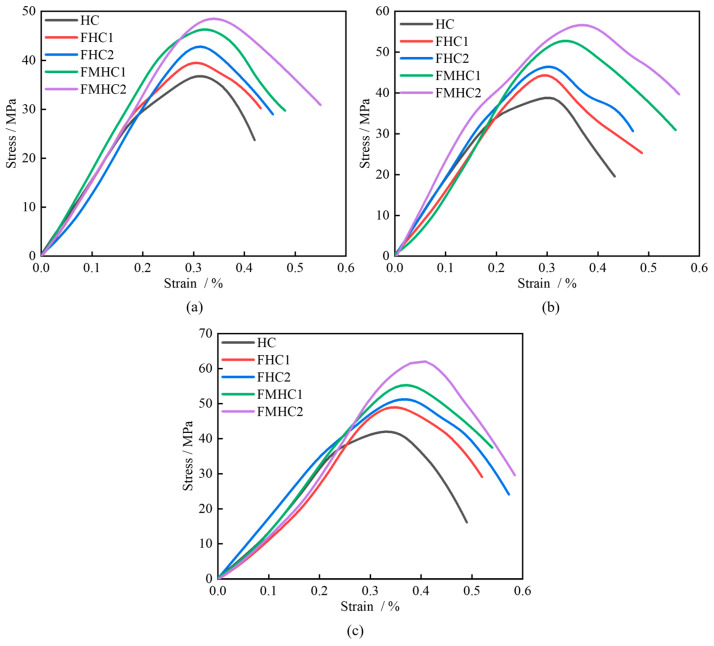
Stress–strain curves at different impact air pressures. (**a**) 0.3 MPa; (**b**) 0.4 MPa; (**c**) 0.5 MPa.

**Figure 9 materials-18-02883-f009:**
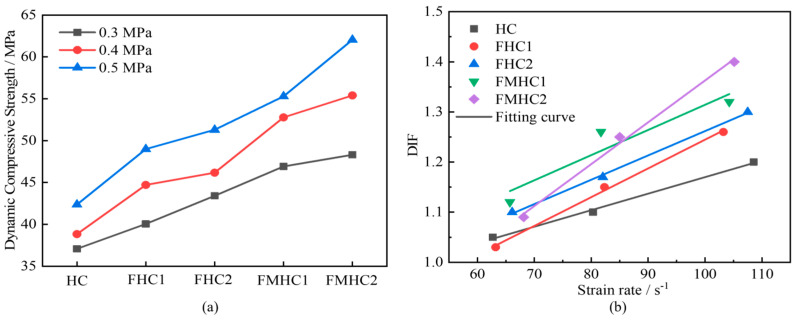
Dynamic compressive strength of different test specimens. (**a**) Dynamic compressive strength; (**b**) dynamic growth factor.

**Figure 10 materials-18-02883-f010:**
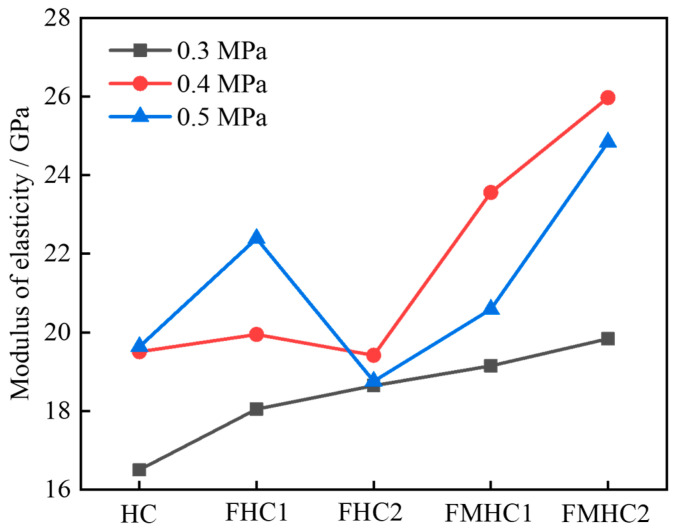
Dynamic modulus of elasticity of different concrete specimens.

**Figure 11 materials-18-02883-f011:**
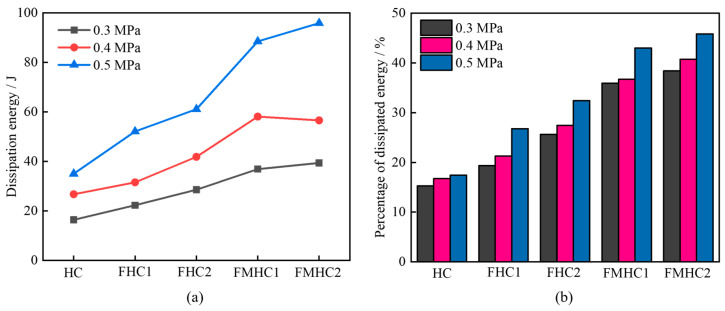
Energy characteristics of different concrete specimens. (**a**) Dissipated energy; (**b**) percentage of dissipated energy.

**Figure 12 materials-18-02883-f012:**
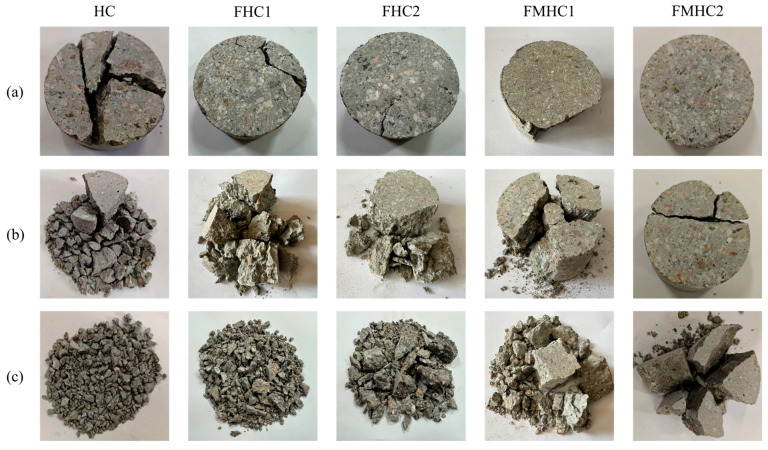
Damage morphology of specimen under impact compression. (**a**) 0.3 MPa; (**b**) 0.4 MPa; (**c**) 0.5 MPa.

**Figure 13 materials-18-02883-f013:**
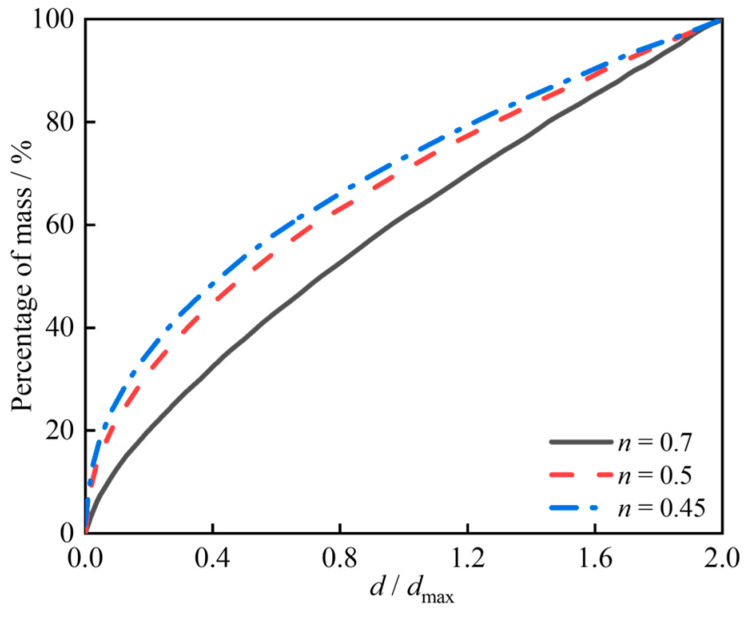
Aggregate grading curve.

**Figure 14 materials-18-02883-f014:**
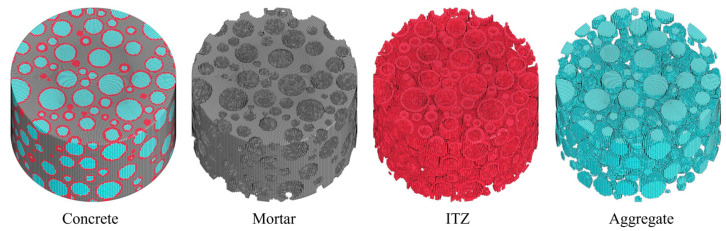
Mesoscale aggregate modeling of concrete.

**Figure 15 materials-18-02883-f015:**
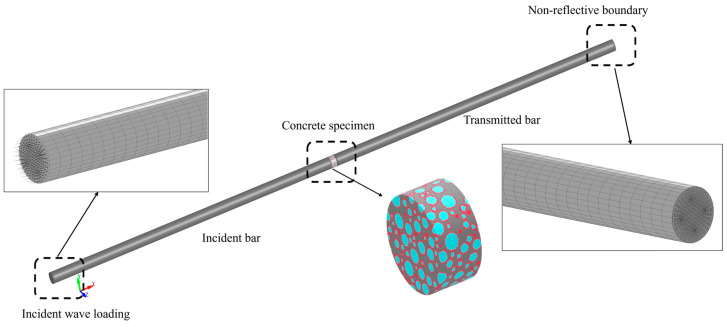
Numerical model of SHPB.

**Figure 16 materials-18-02883-f016:**
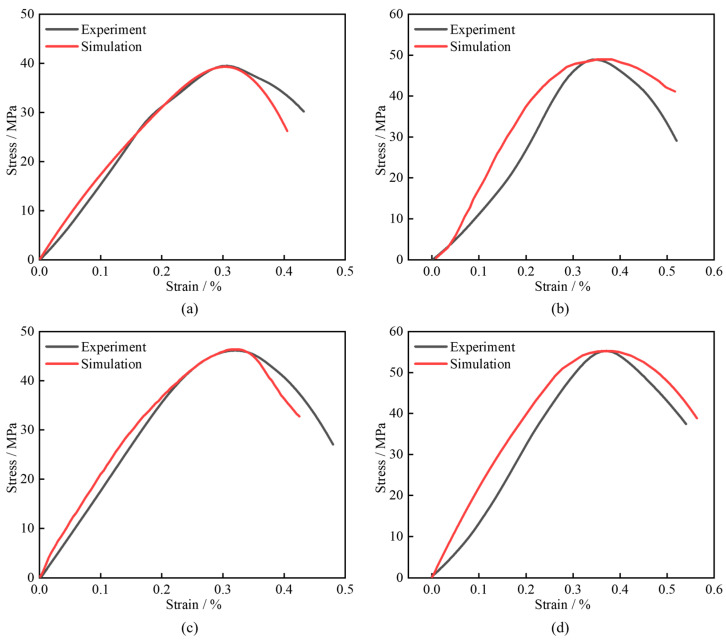
Comparison of the curves between the experiment and the simulation. (**a**) FHC1-0.3 MPa; (**b**) FHC1-0.5 MPa; (**c**) FMHC1-0.3 MPa; (**d**) FMHC1-0.5 MPa.

**Figure 17 materials-18-02883-f017:**
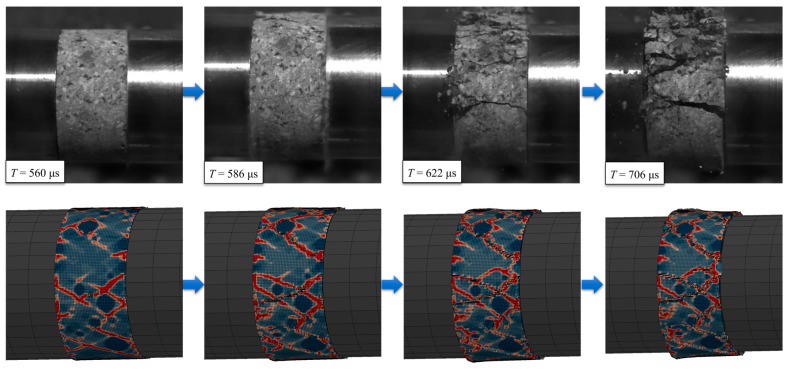
Comparison of dynamic impact compression simulation and experimental damage process.

**Figure 18 materials-18-02883-f018:**
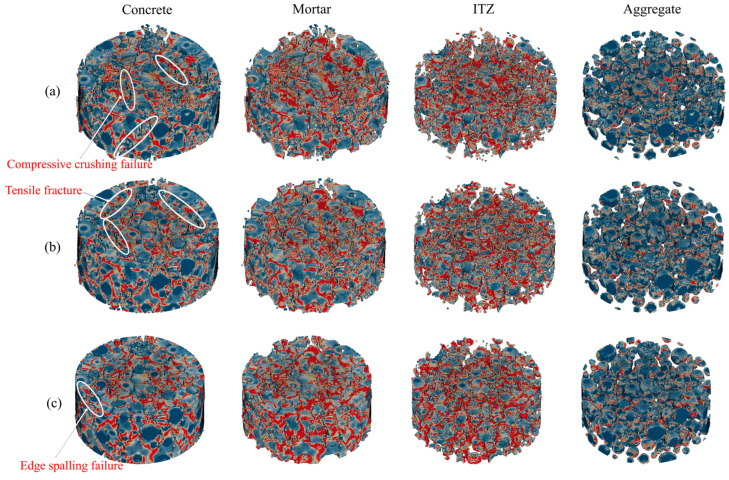
Compression damage patterns of different specimens. (**a**) HC; (**b**) FHC1; (**c**) FMHC1.

**Figure 19 materials-18-02883-f019:**
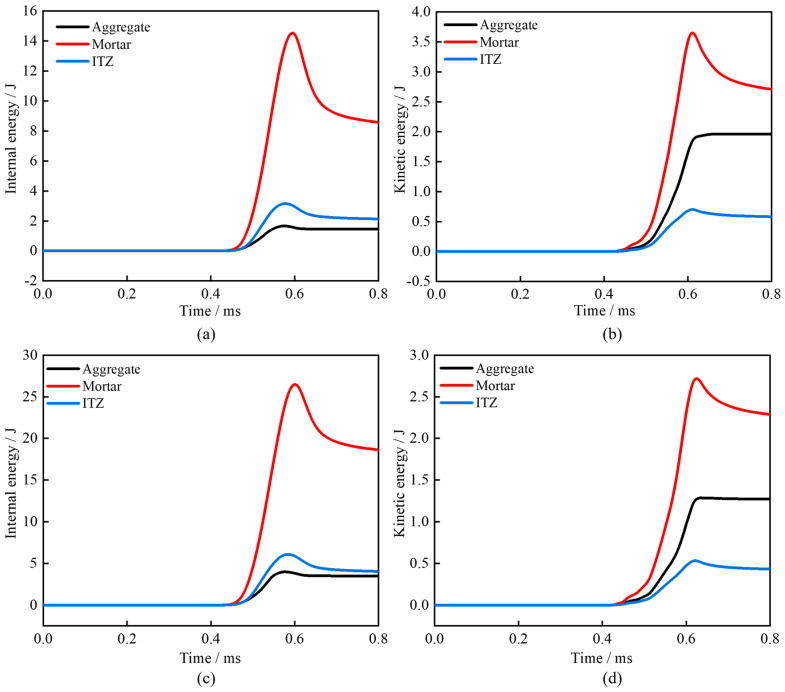
Energy time–history curve. (**a**) FHC1—internal energy; (**b**) FHC1—kinetic energy; (**c**) FMHC1—internal energy; (**d**) FMHC1—kinetic energy.

**Figure 20 materials-18-02883-f020:**
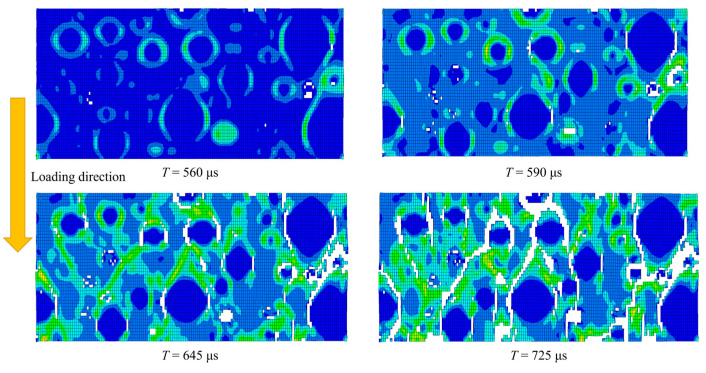
Cross-sectional damage evolution contour map.

**Table 1 materials-18-02883-t001:** Chemical composition of cementitious materials (%).

	SiO_2_	Al_2_O_3_	Fe_2_O_3_	CaO	MgO	TiO_2_	Na_2_O	SO_3_	K_2_O
Cement	23.89	12.22	3.14	49.78	4.10	0.54	0.10	3.54	0.55
Fly ash	40.93	23.53	7.23	20.11	3.21	0.62	0.33	3.98	0.41
Slag powder	33.15	12.28	1.05	40.23	7.75	1.56	0.53	2.71	0.23

**Table 2 materials-18-02883-t002:** Mechanical indexes of coarse aggregates.

Density (kg/m^3^)	Modulus of Elasticity (GPa)	Compressive Strength (MPa)	Tensile Strength (MPa)	Poisson’s Ratio
2660	20.7	92.0	11.5	0.32

**Table 3 materials-18-02883-t003:** Mix designs of concrete (kg/m^3^).

NO.	Water	Cement	Fly Ash	Slag Powder	Fine Aggregate	Coarse Aggregate	Water/Binder Ratio
HC	160	450	0	0	675	1090	0.36
FHC1	160	360	90	0	675	1090	0.36
FHC2	160	270	180	0	675	1090	0.36
FMHC1	160	270	135	45	675	1090	0.36
FMHC2	160	270	90	90	675	1090	0.36

Note: HC stands for high performance concrete, FHC1 stands for high performance concrete with one mineral admixture (20% fly ash) and FMHC1 stands for high performance concrete with two mineral admixtures (30% fly ash and 10% mineral powder).

**Table 4 materials-18-02883-t004:** Energy characteristics of concrete specimens under different strain rates.

Test Group	Strain Rate (s^−1^)	Incident Energy (J)	Dissipated Energy (J)	Percentage of Dissipated Energy (%)
HC-0.3	62.72	107.67	16.42	15.25
FHC1-0.3	63.21	115.26	22.29	19.34
FHC2-0.3	66.11	111.51	28.58	25.63
FMHC1-0.3	65.71	102.67	36.90	35.94
FMHC2-0.3	68.13	102.56	39.38	38.40
HC-0.4	80.32	160.06	26.75	16.71
FHC1-0.4	82.33	148.30	31.54	21.27
FHC2-0.4	82.05	152.61	41.86	27.43
FMHC1-0.4	81.71	158.21	58.11	36.73
FMHC2-0.4	85.02	138.91	56.60	40.75
HC-0.5	108.56	200.79	35.02	17.44
FHC1-0.5	103.24	194.58	52.11	26.78
FHC2-0.5	107.51	188.45	61.13	32.44
FMHC1-0.5	104.26	205.77	88.52	43.02
FMHC2-0.5	105.13	209.13	95.90	45.86

**Table 5 materials-18-02883-t005:** Aggregate RHT material parameters.

Parameters	Value	Parameters	Value
Compressive strength $f_{c}$	92.0 MPa	Failure surface parameter A	1.6
Relative tensile strength $f_{t}^{*}$	0.13	Failure surface parameter N	0.61
Relative shear strength $f_{s}^{*}$	0.30	Tensile and shear meridian $Q_{0}$	0.68
Elastic shear modulus G	8.30 GPa	Lode angle dependence factor B	0.01
Compressive yield surface parameter $g_{c}^{*}$	0.53	Reference compressive strain rate ${\dot{\varepsilon }}_{0}^{c}$	3 × 10^−5^ s^−1^
Tensile yield surface parameter $g_{t}^{*}$	0.7	Reference tensile strain rate ${\dot{\varepsilon }}_{0}^{t}$	3 × 10^−6^ s^−1^
Shear modulus reduction factor ξ	0.50	Critical compressive strain rate ${\dot{\varepsilon }}_{c}$	3 × 10^−19^ s^−1^
Damage parameter $D_{1}$	0.04	Critical tensile strain rate ${\dot{\varepsilon }}_{t}$	3 × 10^−19^ s^−1^
Damage parameter $D_{2}$	1.0	Compressive strain rate exponent $\beta_{c}$	0.015
Minimum damaged strain $\varepsilon_{m}^{p}$	0.01	Tensile strain rate exponent $\beta_{t}$	0.020
Residual surface parameter $A_{f}$	2.24	Crush pressure $p_{\mathrm{el}}$	30 MPa
Residual surface parameter $n_{f}$	0.85	Compaction pressure $p_{\mathrm{co}}$	6 GPa

**Table 6 materials-18-02883-t006:** Basic mechanical parameters of mortar and ITZ.

Component	HC	FHC1	FMHC1
	Quasi-Static Compressive Strength/(MPa)		
Mortar	30.50	33.75	38.24
ITZ	24.40	27.00	30.59

## Data Availability

The original contributions presented in the study are included in the article, further inquiries can be directed to the corresponding authors.

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
