# Peer review of "Dynamic Compressive Behavior and Fracture Mechanisms of Binary Mineral Admixture-Modified Concrete"

_materials, 2025, doi:10.3390/ma18122883_

Round 1
Reviewer 1 Report
Comments and Suggestions for Authors
Thank you very much for reviewing the invitation to the article “Dynamic compressive behavior and fracture mechanisms of bi-nary mineral admixture-modified concrete” submitted to Materials Journal.
After conducting the review process, the following remarks have been raised:
- The language is recommended to be improved. The current language uses highly complex and long sentences that make it hard to follow up.
- Modifying and strengthening the abstract by stating the results quantities and also stating the whole terms before introducing any abbreviations.
- Modifying the keywords
- Modifying and strengthening the introduction by including the eclogical and environmental benefits of using fly ash and authors can use these useful references https://doi.org/10.3390/polym14071423 and https://doi.org/10.1155/2022/4528264.The first paragraph does not have any references; the authors mainly divided the admixture system into two parts, and the second part is not indicated clearly in the introduction. There is a huge mix-up and multiple exchanges between mineral and slag powder; it should be unified to slag powder if the authors mean the same material. The advantage of slag on cement has crucial aspects in enhancing durability; this is recommended to be included in the introduction and can benefit from the following reference https://doi.org/10.1016/j.clet.2024.100770
- The novelty of the study is not clear enough.
- The discussion part is missing from the headlines and in the parts where the authors should link their study with previous studies.

The English language is recommended to be improved. The current language uses highly complex and long sentences that make it hard to follow up.
Author Response
Reviewer #1: Thank you very much for reviewing the invitation to the article “Dynamic compressive behavior and fracture mechanisms of bi-nary mineral admixture-modified concrete” submitted to Materials Journal. After conducting the review process, the following remarks have been raised:
Comments1:The language is recommended to be improved. The current language uses highly complex and long sentences that make it hard to follow up.
Response: We have tried our best to improve the paper and have made some changes. These revisions will not affect the content or framework of the paper. Specific changes have been marked in red in the revised paper. We sincerely thank the editors and reviewers for their hard work and hope that these changes will be recognised.
Comments2:Modifying and strengthening the abstract by stating the results quantities and also stating the whole terms before introducing any abbreviations.
Response: We are grateful to the reviewers for their constructive suggestions. In response, the abstract content of the manuscript was revised accordingly, with particular attention to the following aspects: (1) quantitative experimental results were systematically incorporated to enhance credibility of data; (2) full terms were consistently provided prior to their initial abbreviations throughout the text to ensure terminological clarity.
Comments3:Modifying the keywords
Response: Thanks to your comments, we've changed the keywords to make them more reflective of the article's topic.
Comments4:Modifying and strengthening the introduction by including the eclogical and environmental benefits of using fly ash and authors can use these useful references
https://doi.org/10.3390/polym14071423 and
https://doi.org/10.1155/2022/4528264.The first paragraph does not have any
references; the authors mainly divided the admixture system into two parts, and the
second part is not indicated clearly in the introduction. There is a huge mix-up and
multiple exchanges between mineral and slag powder; it should be unified to slag
powder if the authors mean the same material. The advantage of slag on cement has
crucial aspects in enhancing durability; this is recommended to be included in the
introduction and can benefit from the following reference
https://doi.org/10.1016/j.clet.2024.100770
Response: We appreciate the thoughtful review and constructive feedback provided by the reviewers. We have added a description of the ecological and environmental benefits in the first part of the introduction. And we added a description of the use of fly ash and slag powder as the main object of research in this paper. Relevant references were added for support.
Comments5:The novelty of the study is not clear enough.
Response: We fully accept the comments and have carefully revised the abstract, introduction, and conclusions. The innovation of this paper is emphasized by the comprehensive investigation of the dynamic compressive behavior and fracture mechanisms of binary mineral admixture-modified concrete across multiple scales, including the macroscopic scale (mechanical tests), mesoscopic scale (mesoscale numerical simulations), and microscopic scale (microstructural characterization).
Comments6:The discussion part is missing from the headlines and in the parts where the authors should link their study with previous studies.
Response: We fully accept the comments and have added a discussion section and supplemented the relevance between this study and previous studies in this section.
Reviewer 2 Report
Comments and Suggestions for Authors
The reviewed manuscript, “Dynamic Compressive Behavior and Fracture Mechanisms of Binary Mineral-Admixture-Modified Concrete,” is informative, and the experimental results appear convincing. To strengthen the paper before publication, please address the following minor issues:
- The description for label (c) in Figure 1 is missing; kindly add a clear explanation.
- Redraw or reformat the Figure 15, as the current version does not clearly convey the model developed in the study.
- The distinctions among damage patterns in panels (a), (b), and (c) in Figure 18 are difficult to discern. Consider enhancing the image quality and adding explanatory annotations.
- Rewrite the section of conclusions to focus on the main findings, highlight their contribution relative to existing studies, and outline potential directions for future work.
- Please conduct rigorous comparisons with prior studies, clearly articulate the novel findings of this work, and incorporate additional, recently published references.
Author Response
Reviewer #2: The reviewed manuscript, “Dynamic Compressive Behavior and Fracture Mechanisms of Binary Mineral-Admixture-Modified Concrete,” is informative, and the experimental results appear convincing. To strengthen the paper before publication, please address the following minor issues:
Comments1:The description for label (c) in Figure 1 is missing; kindly add a clear explanation.
Response: Thank you for your review, we have added to the description of Figure 1c.
Comments2:Redraw or reformat the Figure 15, as the current version does not clearly convey the model developed in the study.
Response: Thanks to your review, we have redrawn Figure 15 to show more detail so that it clearly conveys the model developed in the study.
Comments3:The distinctions among damage patterns in panels (a), (b), and (c) in Figure 18 are difficult to discern. Consider enhancing the image quality and adding explanatory annotations.
Response: Thanks to your review, we have added explanatory annotations to Figure 18 and improved the quality of the image to produce a 600 DPI format file.
Comments4:Rewrite the section of conclusions to focus on the main findings, highlight their contribution relative to existing studies, and outline potential directions for future work.
Response: Thanks to your comments, we have rewritten the conclusion section and added potential directions for future work.
Comments5:Please conduct rigorous comparisons with prior studies, clearly articulate the novel findings of this work, and incorporate additional, recently published references.
Response: We sincerely appreciate your valuable suggestions. We have added the differences between this paper and previous studies in the Introduction, Discussion and Conclusion sections. And recently published references are added.
Reviewer 3 Report
Comments and Suggestions for Authors
In the article presented the influence od dynamic and static behaviour on utilization of mineral modified concrete. Special studies concerning dynamic and static load of admixture of minerals in concrete material were provided. Laboratory tests using strain gages to monitor dynamic and static compression of concrete heaving additives of minerals can be observed. Please add Section Discussion taking some material from different parts of the study program. In Section Conclusions please add some more detail findings to get more information concerning overview of the experiment.
Author Response
Reviewer #3: In the article presented the influence od dynamic and static behaviour on utilization of mineral modified concrete. Special studies concerning dynamic and static load of admixture of minerals in concrete material were provided. Laboratory tests using strain gages to monitor dynamic and static compression of concrete heaving additives of minerals can be observed. Please add Section Discussion taking some material from different parts of the study program. In Section Conclusions please add some more detail findings to get more information concerning overview of the experiment.
Response: Thank you very much for your valuable comments, we complete to accept and carefully make changes. We have added a discussion section to analyze the potential reasons for the existence of the findings in this paper, and supplemented the relevance of this study to previous studies. And we also revised the conclusion to make it more complete.
Reviewer 4 Report
Comments and Suggestions for Authors
The applied methodology and aim of the study are described well; however, some minor corrections are still needing.
My comments as follows:
Introduction:
- Fly ash is a combustion by-product of coal and/or solid wastes! For volcanic ash, puzzolan is more prober term!
Experimental investigation
- Provide data about used XRD and SEM equipment in the recent study!
- Provide basic information about fly ash and slag. For instance, are these wastes obtained from one power plant and/or commercial source! Furthermore, if it is possible, please provide the particle size of fly ash and slag.
Numerical simulation
- It is better to mention theoretical background on modelling in section 2
I added some corrections and suggestions in the revised MS. Please check them carefully.

Author Response
Reviewer #4: The applied methodology and aim of the study are described well; however, some minor corrections are still needing. My comments as follows:
Comments1:Introduction: Fly ash is a combustion by-product of coal and/or solid wastes! For volcanic ash, puzzolan is more prober term!
Response: Thank you for your comments, we have checked the terminology in the manuscript.
Comments2:Experimental investigation:Provide data about used XRD and SEM equipment in the recent study! Provide basic information about fly ash and slag. For instance, are these wastes obtained from one power plant and/or commercial source! Furthermore, if it is possible, please provide the particle size of fly ash and slag.
Response: Thank you for your comments. We have added data on the XRD and SEM
equipment in the manuscript. Basic information on fly ash and slag has been added.
Comments3:Numerical simulation:It is better to mention theoretical background on modelling in section 2. I added some corrections and suggestions in the revised MS. Please check them carefully.
Response: Thank you very much for the valuable advice you gave, we fully accept it and have made the changes carefully. We have added the research progress about numerical simulation in the introduction section. And added the establishment theory of numerical simulation. We have also checked the comments marked in the manuscript one by one, and thank you very much for your efforts in reviewing the manuscript.
Round 2
Reviewer 1 Report
Comments and Suggestions for Authors
I appreciate the efforts to revise the manuscript and in my opinion, now it is more soundness and coherent.